# Neuroprotective Effects and Therapeutic Potential of the Citrus Flavonoid Hesperetin in Neurodegenerative Diseases

**DOI:** 10.3390/nu14112228

**Published:** 2022-05-26

**Authors:** Jasmine A. Evans, Patricia Mendonca, Karam F. A. Soliman

**Affiliations:** Division of Pharmaceutical Sciences, Institute of Public Health, College of Pharmacy and Pharmaceutical Sciences, Florida A&M University, Tallahassee, FL 32307, USA; jasmine2.evans@famu.edu (J.A.E.); patricia.mendonca@famu.edu (P.M.)

**Keywords:** neurodegeneration, oxidative stress, neuroinflammation, hesperetin, Nrf2

## Abstract

Neurodegenerative disorders affect more than fifty million Americans each year and represent serious health threats as the population ages. Neuroinflammation and oxidative stress are critical in the onset, progression, and pathogenesis of neurodegenerative diseases such as Alzheimer’s (AD), Parkinson’s (PD), and amyotrophic lateral sclerosis (ALS). A wide range of natural compounds has been investigated because of their antioxidant, anti-inflammatory, and neuroprotective properties. The citrus flavonoid hesperetin (HPT), an aglycone of hesperidin found in oranges, mandarins, and lemons, has been extensively reported to exert neuroprotective effects in experimental models of neurogenerative diseases. This review has compiled multiple studies on HPT in both in vivo and in vitro models to study neurodegeneration. We focused on the modulatory effects of hesperetin on the release of cellular anti-inflammatory and antioxidative stress mediators. Additionally, this review discusses the hesperetin effect in maintaining the levels of microRNA (miRNA) and modulating autophagy as it relates to hesperetin’s protective mechanisms against neurodegeneration. Moreover, this review is focused on providing experimental data for hesperetin’s potential as a neuroprotective compound and discusses reported evidence that HPT crosses the blood–brain barrier. In summary, this review shows the evidence available in the literature to indicate the efficacy of hesperetin in delaying the onset of neurodegenerative diseases.

## 1. Introduction

Neurodegenerative disorders have emerged as a major threat to the aging population and currently affect more than fifty million Americans each year [1]. Neurodegenerative conditions primarily occur in later periods of life, which have become more prevalent because of increased life expectancy. Most common neurodegenerative disorders include Parkinson’s disease (PD), amyotrophic lateral sclerosis (ALS), and Alzheimer’s Disease (AD) [1]. All of which involve memory loss and a decline in neuronal function. The prevalence of these disorders is predicted to gradually increase in the future [2]. Developing treatments for these conditions remains challenging with the forthcoming increase in neurodegenerative disorders [2]. These disorders bring financial and emotional burdens to patients and their caregivers. In 2019, neurodegenerative diseases cost the United States economy 655 billion in healthcare expenses [3,4].

Genetic risk factors play a role in neurodegeneration’s causes, such as the aggregation of toxic proteins and mitochondrial dysfunction [2,3]. For all these disorders, the primary risk factor is age. Other possible components may include oxidative stress, neuroinflammation, vitamin deficiencies, lifestyle behaviors, diabetes, hypertension, stroke, and other preexisting conditions [1]. Environmental factors must also be included in the etiology of neurodegeneration. Exposure to many diverse types of stress leads to cellular homeostatic disruption, which changes normal cellular function, leading to cellular death [5].

Some of the factors that may initiate neurodegeneration are neuroinflammation and oxidative stress. Neuroinflammation is an inflammatory response of the central nervous system (CNS) caused by complex immune responses to injury in the brain [1,2,3]. These injuries lead to the stimulation of glial cells and the secretion of inflammatory cytokines, which cause complications associated with neurodegenerative disorders [6]. Oxidative stress is generated from numerous factors, such as abnormal protein accumulation, high trafficking of calcium ions across the neurons, disturbance in peroxidation, and polyunsaturated fatty acid homeostasis [7]. The activation of multiple biochemical pathways induces damage to proteins, DNA, and lipids, resulting in the death of neuronal cells [8].

Recent findings indicate that natural compounds, such as flavonoids, can potentially prevent cellular injury by attenuating cellular oxidative stress and inflammation [9]. It has been demonstrated that natural products are therapeutic and have few side effects in clinical practice. Historically, flavonoids have been characterized based on their scavenging of free radicals and antioxidant activity. Moreover, it has been indicated that many flavonoids effectively block the neurotoxic pathways associated with neurodegeneration [10,11]. The origin of flavonoid is the Latin root word “flavus”, meaning yellow, because flavonoids are yellow. Flavonoids are naturally occurring polyphenolic metabolites found in fruits, herbs, and vegetables [12]. Additionally, a variety of medicinal and pharmaceutical applications contain flavonoids. These natural compounds can activate enzymes with an antioxidant nature and scavenge reactive oxygen species (ROS) [2,12].

Hesperetin (HPT), a derivative of a naturally occurring flavonoid from the *Citrus* L. plant, has various pharmacological properties, including antioxidative and anti-inflammatory properties [13]. HPT is an aglycone glycoside of hesperidin that widely exists in fruits, vegetables, and traditional Chinese medicinal herbs [14]. HPT has been used to counteract and treat various chronic inflammatory conditions [14,15]. It has also been shown that HPT significantly inhibits interleukin 1 beta (IL-1β) [16]. Moreover, HPT was reported to inhibit inflammation in multiple cell types by controlling the p38 mitogen-activated protein kinase (MAPK) signaling pathway and enhancing the antioxidant protein levels, such as nuclear factor erythroid 2-related factor 2 (Nrf2) in mice brains [15]. This review focuses on the neuroprotective, anti-inflammatory, and antioxidative stress effects of citrus flavonoid HPT on neurodegenerative diseases.

## 2. Neurodegenerative Etiology

Neurodegenerative brain disorders represent a large group that arises from unknown causes and persistently progress. They are characterized as a neurological disorder that affects distinct neurons in specific anatomical areas in heterogeneous groups [17]. The neurodegenerative process is distinguished by the gradual dysfunction and destruction of neurons in susceptible parts of the nervous system [17]. The basic processes that induce neurodegeneration are multifactorial and caused by genetics, endogenous, and environmental factors [18]. These processes include defective protein degradation and aggregation, oxidative stress, mitochondrial dysfunction, interference of neuronal Golgi apparatus, and abnormal ubiquitin–proteasomal system function. These interconnected mechanisms lead to programmed cell death [17,18].

Loss of neurons is the main feature in most neurodegenerative conditions [13]. Genetic and environmental factors participate in variable degrees in the etiology of neurodegeneration. Some neurodegenerative disorders have evidence of familial occurrence, which suggests the cause is genetics, such as diseases with autosomal dominant traits like Huntington’s disease [13]. Less common conditions exist as an autosomal recessive trait, X-linked trait, or a maternally inherited trait [13]. These traits are referred to as genetic neurodegenerative disorders. Others are sporadic, showing small numbers of patients with the disorder being inherited, which is true in AD and PD [13] (Figure 1). Przedbroski et al. [17] characterized neurodegeneration as a neurological disorder that affects distinct neurons in specific anatomical areas in heterogeneous groups.

### Aging and Neurodegeneration

The physical decline, leading to an enhanced threat of disease, is associated with aging, which develops at different rates in different tissues [19]. Among the multiple risk factors for neurodegeneration, aging alone has the most impact. In the elderly population, neurodegenerative disorders are common, and it is rare to find a disease-free brain, especially in very senior individuals [20].

The brain’s aging is an irreversible process that is a critical factor in neurodegenerative diseases [19,20]. The aging process makes patients more prone to neurodegeneration and impairs self-repair abilities [21]. Neurodegeneration, the most prevalent in age-related diseases, indicates the link between neurodegeneration and age-related changes in the brain’s microenvironment, such as epigenetic modification, genomic inability, and the loss of proteostasis [21]. Aging is a major factor for neurodegeneration; however, the exact mechanisms through which aging is associated are not yet identified. Various signaling pathways involved in managing aging include target of rapamycin (TOR) signaling, mitochondrial function, insulin/insulin-like growth factor 1 (IGF-1) signaling, and caloric restriction [22]. Studies recently suggested that the involvement of these pathways may be significant in cognitive decline. Proteins, such as phosphorylated tau, amyloid-β (Aβ) abnormal aggregates, and α-synuclein, have been identified through molecular studies, but it has not been confirmed that they are correlated with cognitive impairment [23].

Recent studies have shown that protein glycation is also involved in forming the amyloid protein. This process gives rise to advanced glycation end product (AGE) formation. Altered AGEs interact with specific receptors, including receptor advanced glycation end products (RAGEs), which are members of the immunoglobulin family and can extensively bind ligands and activate cellular pathways [24]. These receptor products can interact with Aβ, stimulating stress and various signaling pathways within the neuron. AGE and RAGEs have a significant role in AD, either by their incorporation in Aβ plaques and tau tangles or recognition of Aβ and clearance [24,25]. It has been suggested that RAGE induces oxidative stress and inflammation, which may lead to AD [25]. Diet must also be considered along with excessive intake of glycotoxins, which have adverse effects related to the pathogenesis of dementias such as AD and the metabolism of amyloid precursor protein (APP) and regulatory pathways linked to tau phosphorylation [26].

Worldwide, an emerging issue with the increase in life span is the onset of dementia. The commonly occurring form of dementia is AD, which is characterized by memory damage and cognitive injury [21]. The leading factor of AD is aging. The most common pathological findings are senile plaques and neurofibrillary tangles in the brain’s cortex, containing β-amyloid peptides and tau proteins, respectively [19,20]. The collection of misfolded proteins leads to neuronal injury and synaptic damage. Cellular changes in aging may be linked to protein misfolding and aggregation [27]. The aging process is connected to oxidative stress accumulation and mitochondria dysfunction. The brain is vulnerable to malfunctioning mitochondria because of bioenergetic needs [28]. Elevated levels of energetic demands in neural populations in the brain, like degenerate pyramidal neurons in AD, may be affected by the decreasing function of the mitochondria [27,28]. This declining function may impact brain aging and increase the susceptibility of neurons to age-dependent pathological changes [28]. Memory loss is an indicator of AD and normal aging, but the pathology and neurophysiology of both conditions are different. Both histopathological and functional magnetic resonance imaging have exhibited decreased metabolic activity, and neuronal loss begins in the hippocampal region. In the aging brain, a reduced activity occurs first in the subiculum and dentate gyrus of the brain, instead of the hippocampus [29,30,31]. It has been demonstrated that the functional brain imaging that separates brain regions, which interact to accelerate higher-order cognitive function, becomes less coordinated [32]. Alterations in the connectivity of the higher-order brain could be related to myelinated fiber disruption, connecting neurons in different cortical regions or the synaptic physiological changes of aging neurons [32,33,34].

Another age-related neurodegenerative disorder involving protein misfolding is PD. The misfolding of α-synuclein, which develops into Lewy bodies due to its accumulation, starts in the olfactory and subordinate brain stem regions and gradually extends to the midbrain and the cortex [27,35]. Meanwhile, age is the leading risk factor that impacts the onset and occurrence of PD. In a cohort study, patients that had an onset at older ages displayed an accumulation of more Lewy bodies throughout the brain in addition to age-linked plaque pathology [36] (Figure 2).

## 3. The Role of Neuroinflammation in Neurodegeneration

Neuroinflammation is a specific or nonspecific immunological outcome in the CNS induced by microglial activation. In the brain, innate immunity is coordinated by the microglial cells. The inflammatory response participates in the defense against exogenous antigens, but chronic inflammation is implicated in multiple neurodegenerative disorders. Microglia offset disturbances in immunological homeostasis to protect the neurons limited in regenerative capacity [37]. Typically, microglia guard the nervous system by removing debris, demolishing pathogens, and stimulating the immune response. However, when there is a brain injury or neurodegenerative disorder, microglia are activated and release proinflammatory cytokines and neurotoxins [37]. In these instances, released neurotoxins include nitric oxide, IL-1β, interleukin 6 (IL-6), and tumor necrosis factor (TNF-α) [37]. The overproduction of inflammatory mediators may enhance neuronal degeneration. Additionally, several studies have suggested that dominant, nongrowing bacteria contribute to AD, releasing inflammatory elements like lipopolysaccharide (LPS). LPS is found in gram-negative bacteria that activate the immune system, leading to behavioral and memory impairments, ROS generation, and oxidative damage to the brain. In AD, levels of LPS in plasma are increased and are intricately connected with protein aggregation in the brain [38]. Toll-like receptors (TLR) play a critical role in recognizing microbial and bacterial components in the brain, initiating the immune response. A TLR-mediated signaling pathway removes bacteria from the biological system but can damage the brain cells [39]. Microglial cells express toll-like receptor 4 (TLR4) in the brain, which recognizes LPS, and is associated with the release of inflammatory mediators during nuclear factor kappa B (NF-κB) signaling [39].

## 4. Oxidative Stress and Neurodegenerative Diseases

Oxidative stress is an event that is affected by an imbalance in the production and collection of ROS in cells and tissues and the ability of the biological system to remove these ROS products [40]. ROS, as byproducts of oxygen metabolism in normal conditions, play a critical role in multiple physiological conditions. ROS that are frequently elucidated are superoxide radicals (O_2_^−^), hydrogen peroxide (H_2_O_2_), hydroxyl radicals (-OH), and singlet oxygen (^1^O_2_), which are secreted as metabolic byproducts of the biological system. Biological activities, such as the initiation of many transcriptional factors, cell differentiation, immunity, apoptosis, and protein phosphorylation, are dependent on ROS production. However, the presence of ROS inside cells needs to be at low levels [10]. Increased ROS levels show harmful effects on proteins, lipids, and nucleic acids, which are critical in cellular structures. Evidence has shown that oxidative stress may cause the onset or progression of numerous diseases (i.e., cardiovascular disease, diabetes, cancer, and metabolic disorders) [7].

ROS are generally produced via mitochondria during physiological and pathological conditions. Superoxide radicals can be produced by endothelial and inflammatory cells, cellular respiration, lipoxygenases, and cyclooxygenases during arachidonic acid metabolism [41]. Mitochondria have distinctive ROS scavenging capability, but it is not enough to maintain physiological amounts of ROS produced. Excessive ROS in cells is reduced by antioxidant defense mechanisms based on enzymes, such as superoxide dismutase (SOD), catalase (CAT), and glutathione peroxidase (GPx), as protection against ROS-induced damage [42].

Enzymatic responses that bring about ROS are connected to prostaglandin synthesis, phagocytosis, the cytochrome P450 system, and the respiratory chain [43,44]. The generation of superoxide radicals occurs by nicotinamide adenine dinucleotide phosphate (NADPH) oxidase, xanthine oxidase, and peroxidase. Upon formation, superoxide radicals are associated with several reactions that produce hydrogen peroxide, hydroxyl radicals (OH•), hypochlorous acid (HOCl), and many others [40]. Hydrogen peroxide, which is nonradical, produces various oxidase enzymes, such as xanthine and amino acid oxidase. The most reactive among all free radicals produced in vivo is the hydroxyl radical, caused by the Fenton reaction (O_2_^−^ with H_2_O_2_, with Fe^2+^ or Cu^+^). Nitric oxide plays a critical physiological role, and it is synthesized from arginine to citrulline oxidized by nitric oxide synthase [40] (Figure 3).

Studies have shown that oxidative stress has been correlated to many neurological disorders (i.e., AD, ALS, PD, and depression) [45]. In AD, several research investigations have illustrated that oxidative damage has a critical role in the progression of dementia and neuronal loss. AD is responsible for about 80% of dementia cases, which causes a loss of memory, decline in behavioral function, and inefficiency in learning [2,3]. Presently, there is no cure nor therapy aimed at the neurological hallmarks to slow the progression of this neurodegenerative disorder. The etiology of AD has not been elucidated, but pathological characteristics include the aggregation of neurofibrillary tangles and extracellular Aβ plaques [46]. Aβ is a toxic peptide present in the brain of AD patients, which is produced by free radicals and is known to be partly responsible for the onset of neurodegeneration. Several hypotheses have been proposed to explain the AD mechanism, and the most common is the Aβ plaque formation, tau protein destabilization, inflammation, and the cholinergic and oxidative hypothesis [47]. The Aβ peptides seem to be the causative agents in AD, and the development of the Aβ plaques is initiated with the processing of the APP. The processing of APP is internalized and sorted in the endosomes, where APP is processed to generate Aβ [48]. The degradation of Aβ occurs by the endolysosomal pathway or is released by fusing multivesicular bodies into the extracellular space. It should be noted that this is only one operative mechanism for AD pathogenesis mediated by Aβ linked to exosome production, initiating the aggregation process [49].

On the other hand, in ALS, the underlying mechanism is unclear, but diverse cell types (i.e., microglia, astrocytes, macrophages, and T cells) seem to be critical in the pathophysiology of this disease [50]. ALS is a developing disease that can occur sporadically and as a familial disorder. In ROS-mediated oxidative stress ALS, it has been reported that specific oxidative products are elevated both in sporadic and familial ALS model systems [51].

Neuroinflammation and oxidative stress are correlated to the pathogenesis of neurodegenerative conditions. Increased production of ROS leads to oxidative stress, which is coupled with a decrease in antioxidant defense [52,53]. Activated glial and immune cells are the primary producers of ROS and nitrogen species in the pathology of neurodegenerative conditions in the CNS [52]. It is believed that ROS does not cause ALS, but because the cause is unclear, ROS may be likely to exacerbate the progression of the disease [51]. It is believed that oxidative stress may also contribute to ALS by degrading neuromuscular junctions. Mouse models have shown an enhanced sensitivity to the nerve terminal to ROS, promoting a presynaptic decline in neuromuscular junctions [49]. It seems that the over-stimulation of motor neurons in ALS causes the abnormal secretion of acetylcholinesterase, which then decreases the level of acetylcholine in the synaptic cleft. These early-stage changes, paired with inflammatory agents and defective trophic support, lead to neurodegeneration [53].

Studies have also shown that responses to oxidative stress are depressed in ALS. Glutathione, an antioxidant in mammalian cells, is lowered in the motor cortex of patients with ALS compared to healthy ones [54]. Furthermore, the expression of mutant gene TAR DNA-binding protein 43 (TDP-43), which participates in protein production in motor neuron-like cells, increases oxidative stress and mitochondrial damage evoking nuclear deposition of Nrf2, a master regulator of antioxidants, anti-inflammatory, and cytoprotective mechanisms [54]. It has been proven that post-mortem tissues from ALS patients are depleted of Nrf2 mRNA and have low protein levels. Studies in mouse models have shown significant beneficial effects of elevated Nrf2 levels in astrocytes, which are major suppliers of glutathione (GSH) to neighboring neurons [55,56].

Abnormal motor symptoms of Parkinson’s disease, such as rigidity, bradykinesia, resting tremors, and loss of postural reflexes [57], are thought to arise from the loss of substantia nigra dopaminergic neurons in the substantia nigra pars compacta [58]. The exact mechanism remains unclear, but oxidative stress has been considered one of the main pathophysiological mechanisms. Aging is a component coupled with the onset of PD. It is believed that typical cellular dysfunction occurring with aging may cause the increased susceptibility of dopaminergic neurons [58]. Several studies have demonstrated that decreased activity in Complex I of the substantia nigra in the respiratory chain could generate excessive ROS, inducing apoptosis [58].

Antioxidant changes have been reported in PD, even at the initial stages of the disease. Research has shown that GSH levels have been lowered in the substantia nigra pars compacta of PD. However, this finding is not PD-specific [57,59]. In PD, the conditions are aggravated by the presence of ROS-generating enzymes like tyrosine hydroxylase and monoamine oxidase [60]. In addition, dopaminergic neurons are prone to oxidative stress [60]. The nigral dopaminergic neurons contain iron, a catalytic agent for the Fenton reaction, where superoxide radicals and hydrogen peroxide may add to more oxidative stress [61]. There is an intrinsic sensitivity to reactive species, and moderate oxidative stress can prompt a cascade of events leading to cell death [60]. Primary sources of such oxidative stress are generated from the nigral dopaminergic neurons, which are believed to be produced during dopamine metabolism, neuroinflammation, and mitochondrial dysfunction [60]. In the mitochondria, neurons are heavily dependent on the aerobic respiration of adenosine triphosphate (ATP), and hydrogen peroxide and superoxide radicals are normally produced as byproducts during oxidative phosphorylation. Any pathological circumstance that leads to mitochondrial dysfunction can lead to enhanced ROS and overload the cellular antioxidant mechanisms [62]. The presence of oxidative stress affects the peroxidation of specific mitochondrial lipid cardiolipin, which causes a release of cytochrome C into the cytosol, leading to apoptosis [62].

Dopaminergic neurons are intrinsically more ROS-generating, as described previously; anything that prompts further oxidative stress can damage the cell. Mitochondrial Complex I damage in the electron transport causes electrons to leak, which then causes ROS production [63]. Evidence of mitochondrial dysfunction coupled to oxidative stress and dopaminergic cell damage is derived from mutations in the parkin, protein deglycase (DJ-1), and PTEN-induced kinase (PINK) genes, which are mitochondrial proteins coupled with familial forms of PD [64]. Mouse models with the parkin gene absent have shown a reduction in the activity of the striatal respiratory chain along with oxidative damage [64].

## 5. Nrf2 Role in Neurodegeneration and Neuroinflammation

Increased oxidative stress results from changes in antioxidant levels in tissues, enhanced mitochondrial dysfunction, and metal homeostasis changes, where most mitochondria produce additional ROS [65]. Endogenous antioxidant pathways protect cells from oxidative stress by boosting cytoprotective enzyme expression that can scavenge free radicals and decrease cellular injury caused by ROS. Nrf2 controls this pathway by binding to the antioxidant response elements (AREs) at the promoter region of antioxidant genes [66,67].

Nrf2 is a cap ‘n’ collar member of basic leucine zipper proteins, typically bound in the cytosol to the Kelch-like ECH-Associated Protein 1 (Keap1), which targets it and causes it to degrade via the proteasome. In the presence of oxidative stress, nucleophilic cysteine sulfhydryl groups on Keap1 are modified, which results in allosteric conformational changes that reduce the Keap1-dependent degradation of Nrf2 and allow this transcription factor to accumulate in the nucleus [68]. In normal conditions, this is how Nrf2 is activated. However, other mechanisms, including phosphorylation, can result in dissociation from Keap1, increasing nuclear localization [68,69]. In the nucleus, Nrf2 forms a heterodimer with musculoaponeurotic fibrosarcoma (Maf) proteins and binds to the ARE consensus sequence at the promotor of targeted genes [69]. Nrf2 is expressed ubiquitously and, in the brain, is a critical defense mechanism against toxic threats in glial cells and neurons. Nrf2 upregulates multiple antioxidant enzymes, increases the expression of anti-inflammatory mediators, both phase I and II enzymes, and initiates mitochondrial signaling pathways [70,71].

Nrf2 is critical in the crosstalk between antioxidant and anti-inflammatory pathways, which are considered secondary effects to its antioxidant effects. Usually, the nuclear factor kappa-light-chain-enhancer of activated B cells (NFκB) proinflammatory transcription factor is activated by oxidative stress. It can be inhibited by the Nrf2-dependent induction of antioxidant genes, leading to a decrease in the transcription of proinflammatory cytokines [72,73]. However, Nrf2 has been demonstrated to directly control the anti-inflammatory mediator expression of interleukin 17D (IL-17D), G protein-coupled receptor kinase, macrophage receptor with collagenous structure, and platelet glycoprotein 4 (CD36) [74,75]. More recently, Nrf2 has been involved in downregulating the expression of proinflammatory cytokines like TNF-α, IL-6, interleukin 8 (IL-8), and IL-1β in microglia, macrophages, monocytes, and astrocytes [76].

Studies have shown that the expression and activity of Nrf2 are reduced in both aged mice and humans [54,77]. Since oxidative stress, mitochondrial dysfunction, and inflammation are characteristics of the aging brain and aging is the leading risk factor of neurodegeneration, Nrf2 is an appealing target for clinical mediation [65].

### Neuroprotective Effects of Nrf2 Activation

The increased oxidation of proteins and lipids and decreased antioxidant defense have been linked to neurodegenerative diseases. The Nrf2-ARE pathway has been strongly considered a prospective target for preventing these conditions [78,79]. There is less activation of the Nrf2-ARE-dependent gene expression in the CNS in neurons compared to astrocytes [80,81]. Moreover, astrocytes have a greater GSH content than neurons, which is why neurons depend on astrocytes for protection against oxidative stress. Numerous studies indicate that neurons are more resistant to oxidative stress in the presence of astrocytes. Most findings are targeted at astrocytic Nrf2 overexpression-mediated expression, cell-specific overexpression, or the modulation of endogenous neuronal antioxidant capacity via Nrf2 activators [82]. Youdim et al. suggested that Nrf2 knockdown neurons were more susceptible to oxidative stress, while Nrf2 overexpression reversed this effect [83]. Other studies used amyloid protein precursor mouse models with Nrf2 knockout, which showed an increase in oxidative damage; however, the activation or overexpression of Nrf2 protected mice from the toxicity of Aβ [84,85]. Studies using sulforaphane, an Nrf2-activating chemical, showed that treatment after injury could attenuate damage in an Nrf2-dependent manner. Sulforaphane, in one analysis, was administered 30 min after intracerebral hemorrhage and 15 min after traumatic brain injury [86,87]. These studies denote that Nrf2 activation after stress may be favorable for acute toxicity [86]. Evidence suggests that the activation of Nrf2 signaling may offer neuroprotection and offers a potential method for treating neurodegenerative diseases.

## 6. Flavonoids

### 6.1. Classification

Flavonoids are plant-based secondary metabolites that are natural compounds widely found in different medicinal applications. They are polyphenolic compounds with a benzo-γ-pyrone structure responsible for several pharmacological activities [88]. Flavonoids are traditionally classified by oxidation degree, the formation of ring C, and the position at which ring B is connected. Based on their structures, flavonoids are subclassified into flavonols, flavones, isoflavones, and anthocyanidins [89] (Figure 4). Flavonoids have been investigated for their potential health benefits as antioxidants, which scavenge free radical and/or chelate metal ions mediated by functional hydroxyl groups [90]. Antioxidant activity is dependent on conformational changes of functional groups, substitution, configuration, and a total number of hydroxyl groups, which are all critical factors in the mechanisms of antioxidant activity [90,91]. It has been shown that flavonoids can interact with neuronal receptors and modulate signaling pathways, gene and protein expression, and transcription factors. This interaction can control memory and learning processes in the hippocampus [91].

### 6.2. Chemical Structure, Bioavailability, and Dietary Sources

Flavonoids are a large and diverse group of polyphenols with the basic structure of fifteen carbon atoms along with two benzene rings attached by three carbon atoms. Currently, there are more than 5000 flavonoids that have been identified in various plants [89].

Because flavonoids are found in many beverages of plant origin and food, they are called dietary flavonoids. They are the most popular and found in everyday diet, including fruits, vegetables, grains, stems, roots, bark, and flowers. The interest in the use of dietary flavonoids has grown and is focused on addressing oxidative stress-mediated neurodegeneration. The release of flavonoids is by the digestive enzymes in the gastrointestinal tract or physical release via chewing before absorption [89]. Flavonoids are absorbed based on their physicochemical properties, including solubility, molecular size, and their ability to dissolve in the organic phase or polar aqueous phase [93].

### 6.3. Hesperetin

The citrus flavonoid HPT has a plethora of protective properties of interest in neurons and glial cells relative to CNS disorders. HPT is a bioactive compound used traditionally in Chinese medicine with antioxidant, anti-inflammatory, and anticarcinogenic properties [2]. HPT is an aglycone of hesperidin, showing various biological activities (Figure 5). Although hesperidin demonstrates a wide range of biological properties, HPT has greater bioactivity, making it more efficiently absorbed from the gastrointestinal tract [2,15]. Chemically, it is a trihydroxyflavone, which has three hydroxy groups at positions 3,5, and 7. HPT is found abundantly in citrus fruits like oranges and mandarins, which are consumed in the daily diet in the form of juices at 200–590 mg/L [94].

The leading factors that assist in the efficacy of bioactive compounds like HPT [92] include its pharmacokinetics. It has been suggested that after administration, hesperidin is transformed to hesperetin 7-O-glycoside by an enzymatic reaction. The metabolite is hydrolyzed by beta-glucosidase in the small intestine [96]. Human studies have demonstrated that the peak plasma concentration of hesperetin is improved after oral intake of citrus juices, such as grapefruit and orange juices [97]. It has been observed that HPT enters the CNS, where it may exert neuroprotective effects by neutralizing free radicals generated during cellular metabolism [98].

HPT has been found to protect neurons against toxicity and induced oxidative stress, inflammation, and the release of neurotoxic substances in vitro and in vivo neurodegenerative models. Studies have shown in animal models and brain endothelial cells that HPT and other flavonoids are taken up by brain cells [99]. This flavonoid promotes neuronal survival through the phosphatidylinositol 3-kinase and Akt protein kinase B (PI3K-Akt) and MAPK pathways and the recruitment of neuronal progenitor cells affecting astrocytes. HPT demonstrates anti-inflammatory, antioxidant, and neuroprotective effects in various neurodegenerative disorders [15]. It has also been shown that HPT decreases the overexpression of inducible nitric oxide (NO) synthase and proinflammatory cytokines (IL-1β, TNF-α, and IL-6) along with MAPK extracellular regulated kinase 1/2 (ERK1/2) and p38 in LPS stimulated BV-2 murine cell lines [100]. The signaling pathway modulation, antioxidant, and anti-inflammatory activity of flavanones may contribute to observed improvements in cognitive and motor impairment that have been reported by AD animal models treated with HPT [2].

The neuroprotective effects of HPT were investigated in rat models, and the authors determined that HPT and its nanoparticle (at doses of 10 and 20 mg/kg, lasting three weeks) showed significant improvement in learning and cognitive impairment and lowering elevated oxidative stress levels [2]. A similar study was conducted to evaluate the effects of HPT against Aβ-stimulated AD. According to the findings, HPT significantly decreased oxidative stress-mediated neuroinflammation, apoptosis, and neurodegeneration. The study targeted endogenous antioxidant mechanisms, such as TLR4-mediated glial cells neuroinflammation and neurodegeneration. Additional findings showed that HPT reduced cognitive and memory loss in mice [10]. Since neuroinflammation is the primary initiator of AD, inflammation-mediated neurodegenerative disease models have been widely used in different experiments. To further confirm the findings, Khan et al. [2] used LPS stimulated neurodegenerative mouse models of AD to evaluate HPT effects. The results supported HPT’s ability to reduce LPS-induced neurodegeneration and memory loss significantly [2].

To further support the hypothesis that HPT protects against neurodegeneration, multiple studies evaluated the effects on different models of neuronal cell lines of neurodegeneration, which include neuroblastoma SH-SY5Y, PC12 cells, and hippocampal HT22 cells from mice [101]. In the studies performed on neuroblastoma cells, the neuronal damage was stimulated by hydrogen peroxide at a concentration of 75 μmol/L, showing significant protection against peroxide-induced neuronal loss when treated with HPT at a concentration of 0.01 μmol/L. The caspase activity was significantly decreased when HPT was administered. The study also demonstrated an increased expression of ERK1/2 phosphorylation in a dose-dependent manner [60]. When evaluating the PC12 cells, hydrogen peroxide caused cytotoxicity, which promotes cellular damage and reduces antioxidant enzyme production, such as catalase and glutathione peroxidase, changes mitochondrial membrane potential, induces ROS, and diminishes GSH [102]. To analyze the effect of HPT on PD, doses of 50 mg/kg were administered for 1 week. Findings suggested HPT reduced oxidative stress by regulating Nrf2, NF-κB, and apoptotic cell loss. It was also shown that HPT reduced motor dysfunction in PD rat models induced with 6-hydroxydopamine (6-OHDA) [103].

### 6.4. Hesperetin and Microglia

After the immune response is initiated, microglial cells regulate neuroinflammatory responses in the brain. Normally, microglia protect the nervous system by destroying pathogens, eliminating debris, and advancing immune responses. Under physiological conditions, microglia are at rest and serve as host defense and immune surveillance [104]. These cells are vulnerable to changes in their microenvironment and are readily active in response to injury or infection [105]. Once activated, microglia upregulate an array of surface receptors. Microglia also undergo morphological changes from inactive cells to activated and release multiple soluble factors, which are promising in regard to the survival of the neurons [106]. Most of the factors produced, however, are proinflammatory and neurotoxic, including cytokine TNF-α, free radicals such as NO and superoxide, and fatty acid metabolites [105,107,108]. Chronic development of inflammation in the brain can lead to neurodegenerative diseases like AD and PD [37]. HPT’s anti-inflammatory activity in brain-resident glial cells, LPS stimulated astrocytes, and microglial activation was observed in the mouse brain [100]. The astrocyte marker, glial fibrillary acidic protein (GFAP), was significantly increased and the microglial marker, Iba-1, expression was upregulated in the hippocampus of LPS-injected mice compared with control animals [109]. Astrocyte and microglia LPS-induced activation in the hippocampus were significantly decreased by HPT administration [110]. Several findings indicated that HPT suppressed LPS-induced neuroinflammatory response by inhibiting astrocyte and microglial activation [111].

### 6.5. Hesperetin Crossing the Blood–Brain Barrier

There has been an increased concern about the use of dietary flavonoids to combat oxidative stress-induced damage to the CNS and the associated pathophysiological process, such as PD and AD [2,93]. Growing concerns are centered around the entry of flavonoids through the blood–brain barrier (BBB). Furthermore, changes in CNS function may be simply caused by aging, which could exacerbate motor and cognitive modifications [112]. Some studies have analyzed the ability of flavonoids to penetrate through the BBB, which enhanced the findings conducted thus far. The neuroprotective effects of flavonoids and their derivatives against peroxide-stimulated oxidative stress have been reported [113,114]. Although there are demonstrated results on flavonoid-mediated neuroprotection, there is little information on how flavonoids enter the CNS. The metabolism of flavonoids and their method of entry into the systemic circulation after oral absorption has been unclear until recently. There are limited studies on flavonoid neuroprotective properties and their interaction or circulating metabolites on brain endothelial cells, which form the BBB [93]. A layer of endothelial cells forms the BBB, which has a complex identification system for entry into the CNS. This layer of cells is sealed by tight junctions that house a few pinocytotic vesicles and express multiple specific influxes and efflux transporters and metabolic enzymes [93]. These properties alone enable the BBB to regulate the passage of polar molecules [93].

In the case of hesperetin, studies on brain endothelial cells and in vivo show that citrus flavonoids, such as hesperetin, are taken up by brain cells [99]. Youdim et al. [93] demonstrated that, on ECV304/C6 endothelial cell lines, multiple flavonoids, including hesperetin, can enter into isolated mouse and rat brain endothelial cells. It has also been reported that flavonoid aglycones and methylated forms have been uptaken in neurons, suggesting that glucuronidated flavonoids can cross brain endothelium [115]. Youdim et al. [93] showed evidence that derivatives, like hesperetin, cross the BBB by passive transcellular diffusion based on their lipophilicity.

### 6.6. Hesperetin and Neuroinflammation

The main contributor to progressive neurodegeneration is inflammation. In the presence of neuroinflammation, multiple cell types are involved, which include activated microglia and astrocytes [116,117]. The anti-inflammatory effects of HPT have been highlighted in numerous studies conducted. One study analyzed HPT effects against amyloid beta-induced neuroinflammation and neurodegeneration. The findings collectively suggest that HPT lessens Aβ-produced oxidative stress, which reduces the activated microglial cells [118]. Glial cell suppression was coupled with a reduction in NF-κB phosphorylation and the release of inflammatory mediators. Further confirmation of the effects of HPT against Aβ- induced neuroinflammation can be found in an in vitro study demonstrating the inhibition of TLR4 and phosphorylated (p)-NF-κB in comparison to specific pharmacological inhibitors [15]. In a different study, mice were treated with LPS, and the anti-inflammatory effects were evaluated. After the administration of HPT, findings suggest that inflammation was significantly reduced, implied by the lowered expression of TLR4 and p-NF-κB. The comparison of TLR and p-NF-κB with specific pharmacological inhibitors supported the finding that HPT significantly reduced LPS-induced inflammation [15]. Another study demonstrated HPT’s potential anti-inflammatory effects against LPS-activated BV-2 microglial cells [100,101]. In this study, the blocking of MAP kinases was shown with the treatment of HPT, which decreased the expression of p-ERK and p38, reducing inflammatory cytokine levels [119]. Overall, findings exhibit HPT’s potential neuroprotective properties against inflammatory mediators in various neurodegenerative disease models.

### 6.7. Hesperetin and Oxidative Stress

The brain is the most susceptible organ to ROS due to excessive demands for oxygen and the presence of peroxidation-sensitive cells. Extensive research has demonstrated that oxidative stress has a key role in the development of neurodegenerative diseases. Natural and plant-derived compounds like HPT have been found to counteract excessive ROS by scavenging elevated levels and improving endogenous antioxidant defense mechanisms [2]. HPT has demonstrated an upregulation of the expression of Nrf2 and heme oxygenase-1 (HO-1). A previous study showed that HPT has significant protection on RPE-19 cells from increased oxidative stress, inhibiting apoptotic cell death, the production of ROS, and improving the expression of SOD and GSH, which possibly cause the activation of both Keap-1, Nrf2, and HO-1 signaling [120]. Similarly, in the case of potassium oxonate-induced hyperuricemic rats via intraperitoneal injection with or without HPT and orange juice, findings suggest that HPT and orange juice prevented oxidative stress by enhancing antioxidant mechanisms and decreasing lipid peroxidation [121]. To further support HPT’s antioxidative stress properties, a study induced cataracts in rats with sodium selenite and then injected the rats with HPT. Findings here imply HPT and its byproducts lowered oxidative stress in the cataract lens, even though there was a negligible effect on antioxidant levels systemically [122].

Antioxidant effects of HPT have also been evaluated in adult rats treated with lead. Collected findings suggest HPT could reduce the undesirable effects of lead by lowering oxidative damage. These results may be critical in managing lead-induced neurotoxicity [111]. Other studies have shown that cadmium-treated rats administered with HPT reduced lipid peroxidation (LPO) levels, which is a biomarker of thiobarbituric acid reactive substances and lipid hydroperoxides. HPT also improved the levels of plasma nonenzymatic antioxidants, including decreased GSH [123]. Comparable results were collected in mice treated with 7,12-dimethylbenz (a) anthracene (DMBA). HPT distinctly reduced lipid peroxidation levels and protein oxidation and improved the expression of antioxidant defense by improving catalase, SOD, and GSH ratios. HPT was tested in type 1 diabetes mellitus, induced by streptozotocin, showing that the GSH levels and Nrf2 and HO-1, antioxidant genes, were markedly increased with administration [124]. Overall findings propose that HPT is a strong antioxidant flavonoid, reducing increased oxidative stress and damage. In addition, HPT has been found to significantly increase CAT, SOD, and GPx activity in the hippocampal area [125]. It has been previously reported that HPT exerts cytoprotective activity [126].

### 6.8. Hesperetin’s Effect on Canonical and Non-Canonical Mechanisms of Nrf2 Activation

Nrf2 is a critical transcription factor that instructs antioxidant defense genes to maintain cellular homeostasis. Nrf2 regulates the expression of genes involved in the metabolism, immune response, cellular proliferation, and other processes [127]. Nrf2 is considered the “master” regulator against oxidative stress [128]. Normally, Nrf2 remains inactive in the cytoplasm, forming a complex with Keap1, an inhibitory protein. The complex promotes ubiquitination and degradation of Nrf2. Under stress or in the presence of electrophilic compounds, Nrf2 is activated, stimulating the expression of target genes involved in cell protection. This mechanism of activation is known as the canonical mechanism [127].

Nrf2 activation occurs where proteins, such as p62 and p21, disrupt the Keap1/Nrf2 complex and directly interact with Keap1, decreasing the ubiquitination of Nrf2 [129]. Nrf2 disassociates from Keap1 and translocates to the nucleus, where it binds to the ARE in the promoter region of numerous cytoprotective genes, including HO-1 and NADPH [130,131]. This mechanism of activation is known as the noncanonical mechanism. It has been reported that the Nrf2 pathway is closely related to neurodegenerative disorders [130].

In homeostatic conditions, Nrf2 is isolated by the Keap1 complex in the cytoplasm, transferring ubiquitin proteins from E2 ligases to the Neh2 domain in Nrf2 four residue β-hairpin, which interacts with the bottom surface of the Kelch domain of one promotor on the Keap1 homodimer in the open conformation [132,133,134]. The β-hairpin conformation in Nrf2 interacts similarly with the bottom surface of the Kelch domain of the second protomer of the Keap1 homodimer in a second step, called the closed conformation [132]. The Keap1–Nrf2 interaction is also referred to as the hinge and latch model. The Keap1–Nrf2 complex closed conformation allows a lysine-rich α-helix orientation in the Neh2 domain for the Keap1-dependent ubiquitination. This requires the presence of Neddylated-cullin-3 (Cul3) and ring-box 1 (Rbx1) protein [135,136].

Nrf2 inducers oxidize cysteine residues in Keap1, activating a conformational change in the structure. This change in Keap1 induces a Keap1–Nrf2 interaction in a closed conformation. However, in this interaction, lysine residues of Nrf2 are not appropriately positioned for ubiquitination, decreasing Nrf2 degradation. This phenomenon reduces Keap1 levels to isolate a new Nrf2, which is then translocated into the nucleus [137]. Nrf2 heterodimerizes in the nucleus with small Maf proteins with basic leucine zipper (bZip) domains and binds to the ARE sequences, including transcriptional activation [138] (Figure 6). Nrf2 has been reported to heterodimerize with other transcriptional factors like c-Jun, jun dimerization protein 2 (JDP2), and specificity protein 1 (Sp-1), increasing transcriptional activity [139,140]. Additionally, Nrf2 nuclear activity is regulated negatively by cellular master regulator of cell cycle entry and proliferative metabolism (c-Myc) dimerization, which represses genes regulated by Nrf2 [141]. Activation of Nrf2 is ended with its nuclear export through nuclear export signal (NES) sequences in Nrf2. Phosphorylation by glycogen synthase kinase 3 beta (GSK3β) is an essential process in nuclear export [139]. The study of the Nrf2 activation mechanism has been focused on the canonical pathway, with potential therapeutic effects against oxidative stress through Nrf2 signaling. This pathway involves proteins such as p62, p21, and breast cancer type 1 susceptibility protein (BRCA1) [127].

Nrf2 regulates antioxidant responses by transcriptionally activating HO-1 expression. Additionally, Keap1 regulates Nrf2 negatively by inhibiting the activation of Nrf2. HPT was evaluated for its potential to induce Nrf2 activation and degrade Keap1 in association with upregulation of HO-1. RAW 264.7 cells were treated with HPT in a range of concentrations, and a Western blot analysis of the proteins was conducted. Findings indicated that HPT significantly increased total Nrf2 and Keap1 degradation [142,143]. Lui et al. treated myocardial ischemic mice induced with isoproterenol (ISO). There were separately treated mice groups with low doses and high doses with HPT combined. Findings suggest that HPT protects ISO-induced myocardial ischemia by modulating oxidative stress, apoptosis, and inflammation by Nrf2 activation [144].

### 6.9. Possible Effects of Hesperetin on Autophagy Modulation

Autophagy is a process found in low levels under normal conditions known as self-eating, which maintains cellular homeostasis balancing catabolic mechanisms. Autophagy is controlled by numerous pathways targeting multiple disease states, including viral and bacterial infections, cancer, diabetes, and neurodegenerative diseases [146]. It is activated in response to stress to remove excess damaging cellular proteins and organelles. Autophagy comprises five key steps: induction, nucleation, vesicle lengthening and maturation, vesicle fusion, and degradation and recycling [146]. These processes are regulated by the recruitment of autophagy-related protein repressed by the mechanistic target of rapamycin complex 1 [146].

Claims that autophagy deficiency promotes neurodegeneration are strengthened by studies that uncover a class of diseases caused by mutations in critical autophagic genes. β-propeller protein-associated neurodegeneration (BPAN) is a type of neurodegeneration introduced with a movement disorder and declines in intellect [147]. This disease is an X-linked dominant with the loss of function mutation in WD Repeat Domain 45 (WDR45)/WD repeat domain phosphoinositide-interacting protein 4 (WIPI4), a β-propeller scaffold protein that is an autophagic gene. In mice, autophagy in the brain was further highlighted in studies showing that the neuron-specific loss of critical autophagic proteins leads to neurodegenerative phenotypes in the absence of contributing factors [148,149]. Autophagy is a critical regulator of intracytoplasmic aggregate-prone protein levels that cause neurodegenerative diseases, including ALS, PD, HD, and several dementias [150,151,152,153]. When autophagy is impaired, there is a delay in removing said aggregates, and self-eating is initiated when stimulated [147].

It has been reported in several neurodegenerative diseases that autophagic dysfunction contributes to these disorders. It was first suggested that autophagy alteration in AD was caused by accumulating autophagic vesicles in impacted neurons. Initially considered, increased autophagy was a representation of AD, but, more recently, findings imply that vesicle accumulation is because of reduced autophagic clearance [154,155]. The γ-secretase complex is required for Aβ production. Presenilin-1 (PS1) is part of complex functions that facilitate the N-glycosylation of V0 sector isoform a1 (V_0_a_1_) subunit lysosomal v-ATPase and traffic the enabling acidification of the lysosome [156]. The mutation of PS1 and presenilin-2 (PS2) causes familial autosomal-dominant AD, resulting in neuronal loss, amyloid deposition, and lysosomal pathology [157,158]. Lysosome acidification loss and function lead to autophagosome accumulation. It has also been shown that decreasing the expression of beclin 1 (BECN1), the autophagic gene coding for beclin1, reduced mRNA levels in neuronal autophagy of AD brain tissue [159]. BECN1 can be cleaved by caspase3, which may be activated in AD neurons, causing the impaired formation of autophagosomes [160]. In a haploinsufficient mice model, overexpressing amyloid precursor protein crossed with beclin1 demonstrated autophagy disruption and heightened pathology [147]. The loss of beclin1 activity decreases autophagosome formation, which may be coupled with defects in autophagosome biogenesis and degradation, resulting in defective lysosomes [147]. Mutations in the leucine-rich repeat kinase 2 (LRRK2) gene in autosomal dominant PD have been indicated to regulate autophagy negatively following LRRK2 knockdown [161]. Another form of PD, the Kufor–Rakeb syndrome, is linked to mutations in ATPase cation transporting 13A2 (ATPase 13A2), encoding for lysosomal ATPase. This ATPase type regulates lysosomal acidification, which is needed for autophagosome–lysosome fusion and substrate degradation [162]. Many studies have implied that Aβ influences neurodegenerative diseases; however, Aβ is a substrate of autophagy.

Pucci et al. demonstrated that in methylglyoxal-treated mice, PS1 was significantly increased based on a high intake of AGE activators in the diet, which increased oxidative stress in AD mice models, suggesting a link between RAGE and PS1 as it relates to the worsening of dementia, such as AD [163]. Recent studies have demonstrated the importance of AGEs in neurodegenerative conditions with the abnormal accumulation of glycated proteins in the brain. Furthermore, as previously stated, the diet can increase AGE collection leading to increased oxidative stress and inhibition of Nrf2, both of which progress to neurodegenerative diseases. Mastrocola et al. [164] demonstrated that mice fed with fructose treated with pyridoxamine displayed an inhibited production of AGEs and upregulated Nrf2. Evidence also shows that increased levels of RAGEs can trigger the stimulation of NFkB, leading to decreased expression of Nrf2 [165].

Recently, flavonoids have been investigated for their anti-inflammatory and antitumor properties and their connection to autophagy [166,167]. Some beneficial effects of some flavonoids in oncology pathology are commonly coupled with their activity on autophagy and nonapoptotic cell death [168]. Multiple articles showed that flavonoids could induce autophagy both in vivo and in vitro, resulting from their antioxidant activity in proteosomes and mitochondrial endoplasmic reticulum [169].

Flavonoids have protective effects against bacterial and viral infections that promote the activation of autophagy like baicalin, epigallocatechin gallate (EGCG), naringenin, and quercetin [170,171]. Endothelial cells treated with a high glucose concentration demonstrated that quercetin induces autophagy by the negative regulation of p62 and beclin-1 and microtubule-associated protein light chain 3 lipidated limestone calcined clay cement (LC3-II) activation [172]. Oxidative damage is reduced and there is an improvement in antioxidant defense. This suggests that flavonoids could be potential candidates in treating neurodegeneration linked with defective autophagy, demonstrated by their ability to modulate autophagy [172]. Quercetin in AD has been shown to reduce abnormal protein aggregates such as β-amyloid peptides and hyperphosphorylated tau protein through the autophagy pathway [173,174]. In primary rat neuron culture, EGCG decreases phosphorylated tau protein by upregulation of mRNA expression of autophagy adapter proteins [175]. An in vivo PD model showed that quercetin and baicalin amplify autophagic functions, and neurotoxicity induced by rotenone was improved [173,176]. Autophagy plays a protective role in pathophysiology, and it should be noted that the modulation of autophagy is a critical target in regulating metabolic diseases [177].

It was described that HPT affects A1-42-induced autophagy with the impairment of insulin-stimulated glucose uptake in Neuro-2A cells, indicating that this phenomenon may be related to cellular autophagy [126]. Findings in this study suggest that HPT may be a potential therapeutic agent to prevent the progression of neurodegeneration by modulating autophagy [126].

### 6.10. The Possible Protective Effects of Hesperetin on MicroRNA

MicroRNA (miRNA) degradation is emerging as a contributor to neurodegeneration, influencing most of the mechanisms responsible for neurodegenerative diseases [178]. It is said that neurodegeneration can be considered an RNA disorder where miRNA plays a critical role [179]. The miRNA could provide therapeutic targets based on suggesting that one miRNA can affect several target genes, potentially modifying an entire disease phenotype by modulating one molecule. The miRNAs are single-stranded RNA molecules that inhibit gene expression by complementary binding sequences in the three prime untranslated region (3′UTR) of target genes [180]. In the brain, dopaminergic neurons are dependent on the functionality of the miRNA network, which has been observed in vitro and in vivo [181]. In neurodegeneration, miRNA has been described as regulating APP expression [182]. Recent research shows that miRNAs may induce TLR and mediate neuroinflammatory processes [183].

Various preclinical studies have indicated their possible therapeutic potential to alter miRNAs levels by different mechanisms. The quercetin intake effect is mediated primarily by its ability to influence miRNA expression. The miRNA has a regulatory role in major processes, including inflammation, apoptosis, proliferation, differentiation, neurodegeneration, and immune response. In neurodegenerative diseases like AD, quercetin has shown a modified regulation of miRNA [184]. In the various stages of AD, multiple cells affect many miRNAs, potentially those involved in AD. An in vitro study using PC-12 neuronal cells demonstrated that quercetin markedly protected them from hydrogen peroxide-induced cellular death [185,186]. ROS and neurodegenerative processes are linked and act as the regulators and effectors of miRNA and their target proteins [187,188]. Over 135 miRNAs are found in neuronal cells in response to oxidative stress, and there was a reduction in the number of miRNAs identified as critical in antioxidant responses triggered by quercetin [181]. Neuronal cells pretreated with quercetin showed that 14 miRNAs had altered expression induced by oxidative stress; miRNAs observed here were linked to the antioxidant ability of quercetin [188]. In diabetes-induced memory impairment mice, quercetin has been demonstrated to increase the expression of miR-146a, miR-9, and precursor proteins. A critical finding in this study was that quercetin normalizes gene expression in the hippocampus of diabetic rats, modulating pathological inflammation. It was suggested that miRNA-dependent anti-inflammatory mechanisms justify the neuroprotective effects of quercetin [189]. An AD mouse model with vitamin D deficiency supplemented with quercetin had a significant reduction in miRNA levels, suggesting that tau amelioration upon treatment with quercetin may modulate miRNA expression. Modifications of miRNA were observed, which indicates that quercetin treatment causes miRNA differential expression in specific regions of the brain that can participate in neurodegenerative disease, particularly AD [190].

Although there is little knowledge regarding this, EGCG has been demonstrated to upregulate miRNA expression in chondrocytes and decrease inflammation in osteoarthritis via miRNA-199a-3p, reducing cyclooxygenase-2 (COX2) stimulation [191]. Both prostaglandin E2 and interleukin 1 were inhibited. It was also shown that EGCG reduces the prevalence of miRNA in the serum in APP/PSI transgenic mice [192]. These findings suggest that EGCG may indirectly affect miRNA, decreasing age-related neuroinflammation [183].

Dong et al. investigated the protective effects and mechanism of HPT in the lungs of mice. Results on injured lung tissue showed a significant increase in miRNA levels in acute lung injury mice after HPT treatment, suggesting that HPT has a protective effect on miRNA [193]. These findings indicate that HTP may be a potential agent for increasing miRNA expression [193].

## 7. Conclusions

Neurodegenerative diseases can be triggered by aging, elevated oxidative stress, neuroinflammation, and environmental factors. However, outcomes of neurodegeneration may be attenuated with the use of different flavonoids. HPT is a flavonoid with anti-inflammatory effects through the downregulation of inflammatory cytokines and antioxidative stress properties by the activation of the Nrf2/Keap1 pathway and the subsequent induction of antioxidant genes transcription. Reports show evidence that HPT can cross the BBB. Moreover, HPT was demonstrated to modulate autophagy and increase miRNA levels, making it a potential candidate for treating neurodegenerative diseases or delaying the onset of the disease. The available literature allows the conclusion that HPT may be a potential candidate for the management of neurodegenerative disease, offering neuroprotection, although further investigations are still needed (Figure 7).

## Figures and Tables

**Figure 1 nutrients-14-02228-f001:**
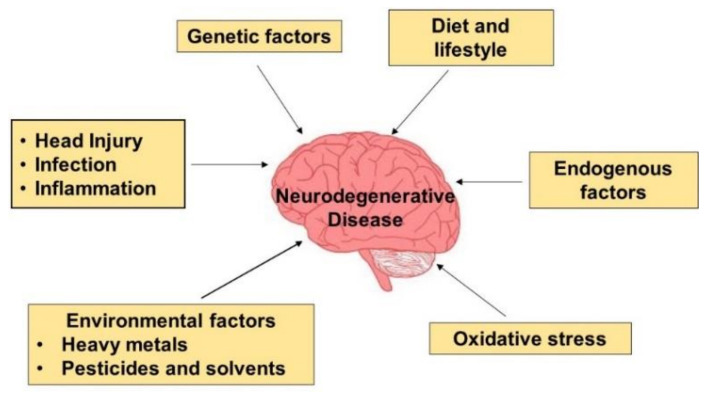
Etiology of neurodegenerative diseases. In most neurodegenerative diseases, the onset of the disease is unknown. Environmental, genetic, and endogenous factors and oxidative stress must be considered when investigating the mechanism of the onset of neurodegeneration.

**Figure 2 nutrients-14-02228-f002:**
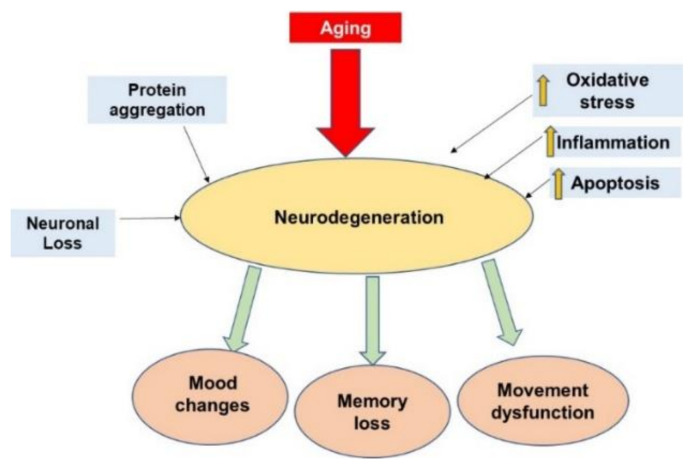
Common factors that induce neurodegeneration. Aging is one of the main factors associated with neurodegeneration, leading to changes in mood, memory loss, and movement dysfunction. Moreover, protein aggregation, neuronal loss, inflammation, oxidative stress, and apoptosis may also promote neuronal death. Yellow arrows indicate an increase.

**Figure 3 nutrients-14-02228-f003:**
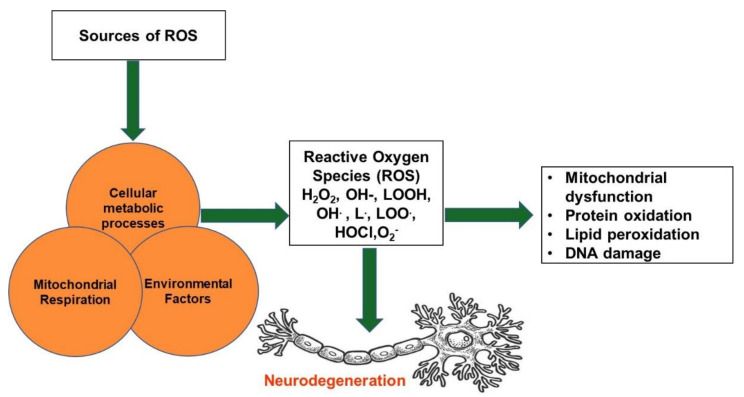
Oxidative stress and its effects on neurodegeneration. The introduction of various sources led to the production of reactive oxygen species causing neurodegeneration, which leads to DNA damage and mitochondrial dysfunction. H_2_O_2_, hydrogen peroxide; OH-, hydroxide; LOOH, lipid hydroperoxides; OH•, hydroxyl radical; L•, lipid radical; LOO•, peroxy radical; HOCl, hypochlorous acid; O_2_^−^, superoxide radicals.

**Figure 4 nutrients-14-02228-f004:**
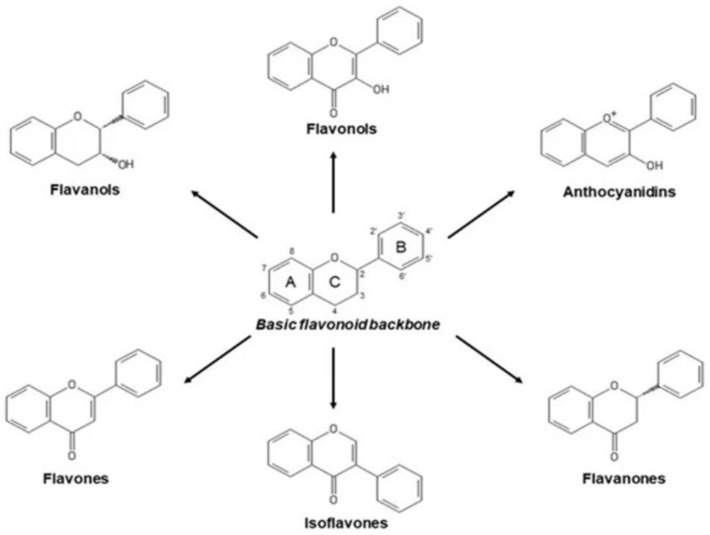
Basic chemical structure of the types of flavonoids. Three rings are considered the backbone of the flavonoid. The oxygen atom is in the first position in the heterocyclic ring labeled as C [92]. O, oxygen; O^+^, oxygen ion; OH, hydroxyl group; A, C, B, rings structure.

**Figure 5 nutrients-14-02228-f005:**
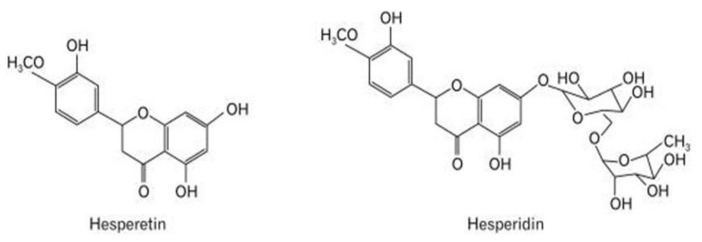
Chemical structure of hesperidin and hesperetin [95]. H_3_CO, methoxy group; CH_3_, methyl group.

**Figure 6 nutrients-14-02228-f006:**
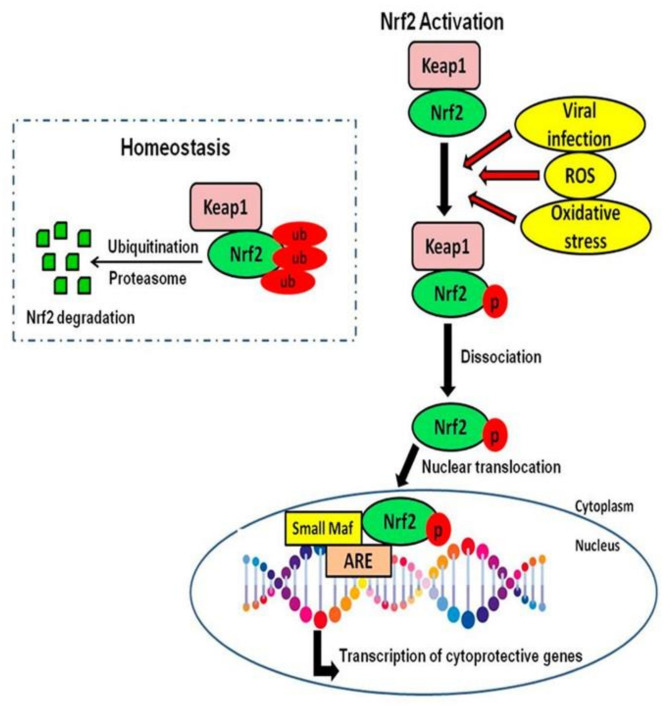
Nrf2 mechanism of action. Oxidative stress induces Nrf2 modification, leading to the release, stabilization, and Nrf2 nuclear translocation. Nrf2 binds to the ARE in the promoter region, transcribing multiple antioxidant genes [145]. Nrf2, nuclear factor-erythroid factor 2-related factor 2; Keap 1, Kelch-like ECH-associated protein 1; ub, ubiquinated; p, phosphorylation; Maf, musculoaponeurotic fibrosarcoma; ARE, antioxidant response element.

**Figure 7 nutrients-14-02228-f007:**
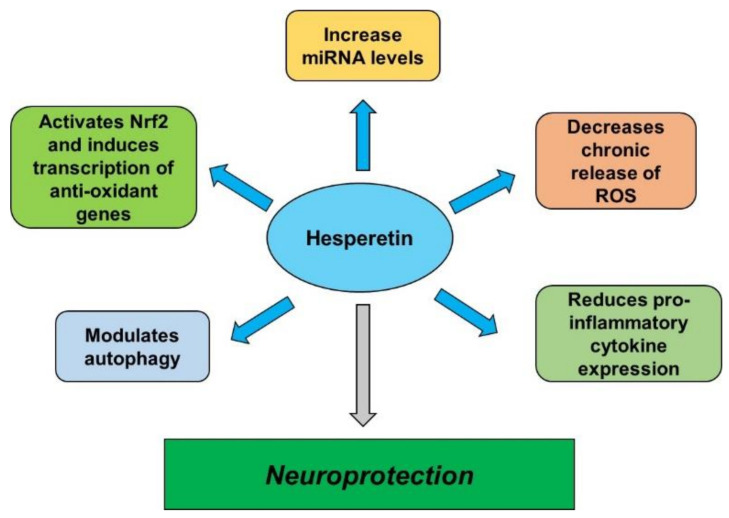
Hesperetin’s neuroprotective effects. Studies show that in vivo and in vitro administration of hesperetin increases miRNA levels, decreases the expression of proinflammatory cytokines, reduces autophagy, and reduces the chronic production of ROS, indicating a neuroprotective mechanism for this compound. miRNA, microRNA.

## Data Availability

Not applicable.

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
