# Peer review of "Neuroprotective Effects and Therapeutic Potential of the Citrus Flavonoid Hesperetin in Neurodegenerative Diseases"

_nutrients, 2022, doi:10.3390/nu14112228_

Round 1
Reviewer 1 Report
The review entitled " Neuroprotective Effects and Therapeutic Potential of the Citrus Flavonoid Hesperetin in Neurodegenerative Diseases" by Jasmine Evans, Patricia Mendonca and Karam F.A. Soliman presents as a thoroughly structured, coherent piece of work. The review is not only valuable for dedicated scientists, likely it is also useful for interested readers and students not so familiar with molecular mechanisms leading to neurodegeneration and/or oxidative stress. If anything to suggest, I would kindly ask the authors to increase the letter-size in their nice illustrations (2,3,6), maybe populate box “ROS” in Figure 3 with ROS-species (O2-,H2O2,-OH,-O2, OH*, LOOH, L*, LOO*, HOCl).
Author Response
Dear editor:
We are pleased to submit the revised version of the Manuscript ID: nutrients-1719825. Title: "Neuroprotective Effects and Therapeutic Potential of Citrus Flavonoid Hesperetin in Neurodegenerative Diseases." We appreciate the constructive criticisms and comments of the reviewers, and we have addressed each of their concerns as outlined below.
Reviewer 1:
1) If there is anything to suggest, I would kindly ask the authors to increase the letter size on their nice illustrations (2,3,6) and maybe populate the box "ROS" in figure 3 with ROS-species.
Response: We appreciate and accept the reviewer's suggestions. The letter size of all figures was increased, and the additions to figure 3 have been completed. A new figure replaced the original figure 6.

Reviewer 2 Report
In the manuscript authored by Jasmine Evans et al. authors summarized the potential neuroprotective and therapeutic effects of the citrus flavonoid hesperetin for neurodegenerative diseases such as Parkinson’s and Alzheimer’s disease.
I just have some comments:
-I suggest authors to include in the aging process or in Neurodegenerative Etiology the role of advanced glycation end products, RAGE and glycotoxins the contributing role in exacerbating dementia. The RAGE receptor is also intertwined with Nrf2 and NFkB pathways. Recently, a link between RAGE and PS-1 has also been demonstrated.
-In the introduction references are missing.
-Authors should also include in figures diet and lifestyle.
-Figures style can be highly improved. But among all, I strongly suggest modifying Figure 6 . Authors can also ask for authorisation to report figures already published in other articles as the Nrf2 and Keap 1 transcriptional activation is a wide use and common representation that can be found in the literature.
Author Response
Dear editor:
We are pleased to submit the revised version of the Manuscript ID: nutrients-1719825. Title: "Neuroprotective Effects and Therapeutic Potential of Citrus Flavonoid Hesperetin in Neurodegenerative Diseases." We appreciate the constructive criticisms and comments of the reviewers, and we have addressed each of their concerns as outlined below.
Reviewer 2:
1) I suggest authors to include in the aging process or in Neurodegenerative Etiology the role of advanced glycation end products, RAGE, and glycotoxins in the contributing role in exacerbating dementia. The RAGE receptor is also intertwined with Nrf2 and NFkB pathways. Recently, a link between RAGE and PS-1 has also been demonstrated.
Response: We appreciate and accept the reviewer's suggestions. As advised, these topics were added. We included the role of advanced glycation end products and RAGE and glycotoxins contributing roles in dementia in section 2.1 Aging and Neurodegeneration. (paragraph 3, lines 1-12).
We also included the link between RAGE receptor and Nrf2 and NF-kB, and RAGE and PS1 in section 6.9 Possible Effects of Hesperetin on Autophagy Modulation. (paragraph 4, lines:1-11).
2) In the introduction, references are missing.
Response: We have revised the introduction, and references have been included.
3) Authors should also include in figures diet and lifestyle
Response: All figures have been revised, and diet and lifestyle have been added to figure 1 as we are discussing the etiology that may lead to neurodegeneration.
4) Figures' style can be highly improved. But among all, I strongly suggest modifying Figure 6. Authors can also ask for authorization to report figures already published in other articles, such as the Nrf2 and Keap 1 transcriptional activation is a wide use and common representation that can be found in the literature.
Response: As suggested by the reviewer, figure 6 has been replaced.

Round 2
Reviewer 2 Report
Authors modified the manuscript according to the reviewer's suggestions. the manuscript has been extensively revised and it has been highly improved.